# Escalation Time to Open Triple Combination Therapy from the Initiation of LAMA versus ICS/LABA in COPD Management: Findings from Comparing the Incidence of Tiotropium and ICS/LABA in Real-World Use in South Korea (CITRUS) Study

**DOI:** 10.3390/jpm11121325

**Published:** 2021-12-07

**Authors:** Ye Jin Lee, Chin Kook Rhee, Yong Il Hwang, Kwang Ha Yoo, So Eun Lee, Doik Lee, Yong Bum Park, Youlim Kim

**Affiliations:** 1Division of Pulmonary and Critical Care Medicine, Department of Internal Medicine, Seoul National University Hospital, Seoul 03080, Korea; pulmoyjlee@daum.net; 2Division of Pulmonary, Allergy and Critical Care Medicine, Department of Internal Medicine, Seoul St. Mary’s Hospital, College of Medicine, The Catholic University of Korea, Seoul 06591, Korea; chinkook77@gmail.com; 3Division of Pulmonary, Allergy and Critical Care Medicine, Department of Internal Medicine, Hallym University Sacred Heart Hospital, Anyang-si 14068, Korea; hyicyk@gmail.com; 4Division of Pulmonary, Allergy and Critical Care Medicine, Department of Internal Medicine, Konkuk University Hospital, School of Medicine, Konkuk University, Seoul 05030, Korea; 20010025@kuh.ac.kr; 5Medical Affairs, Boehringer-Ingelheim Korea, Seoul 04527, Korea; soeun.lee@boehringer-ingelheim.com; 6Real-World Solutions, IQVIA, Seoul 05510, Korea; doik.lee2@iqvia.com; 7Division of Pulmonary, Allergy and Critical Care Medicine, Department of Internal Medicine, Kangdong Sacred Heart Hospital, Hallym University College of Medicine, Seoul 05355, Korea; bfspark2@gmail.com

**Keywords:** chronic obstructive pulmonary disease, inhaled corticosteroids, long-acting β2-agonists, long-acting muscarinic receptor antagonist

## Abstract

Background: bronchodilators are the key treatment for chronic obstructive pulmonary disease (COPD), however, inhaled corticosteroids (ICSs)/long-acting β2-agonists (LABA) are widely prescribed. We compared the escalation time to open triple combination therapy between long-acting muscarinic receptor antagonists (LAMA) and ICS/LABA in COPD management. Methods: this retrospective study included COPD patients selected from the National Health Insurance Service of South Korea from January 2005 to April 2015. The primary outcome was the escalation time to triple therapy in patients who initially received LAMA or ICS/LABA. Other outcomes included risk factors predisposing escalation to triple combination therapy. Results: a total of 2444 patients were assigned to the LAMA or ICS/LABA groups. The incidences of triple combination therapy in the LAMA and ICS/LABA groups were 81.0 and 139.8 per 1000 person-years, respectively (*p* < 0.001); the median times to triple therapy escalation were 281 and 207 days, respectively (*p* = 0.03). Treatment with ICS/LABA showed a higher risk of triple therapy escalation compared to LAMA (hazard ratio (HR), 1.601; 95% confidence interval (CI), 1.402–1.829). The associated risk factor was male sex. (HR, 1.564; 95% CI, 1.352–1.809). Conclusions: the initiation of COPD treatment with LAMA is associated with a reduced escalation time to triple therapy compared with ICS/LABA.

## 1. Introduction

Bronchodilator therapy, including a long-acting β2-agonist (LABA) or a long-acting muscarinic receptor antagonist (LAMA), is the key treatment for chronic obstructive pulmonary disease (COPD) [1,2]. According to recent COPD guidelines [1], even in GOLD grade 4, group D patients, LAMA monotherapy may be sufficient for disease control. However, inhaled corticosteroids (ICS) plus LABA (ICS/LABA) continue to be widely prescribed in clinical practice [3]. For patients with a blood eosinophil count of >300/μL or frequent exacerbations of COPD, ICS has a clinical role in COPD management, and several studies have confirmed that ICS/LABA is more effective than LAMA for exacerbation prevention [4,5,6,7]. However, side effects such as pneumonia may occur in ICS-treated patients [7,8,9].

When patients with COPD experience frequent exacerbations, reduced lung function or worsening symptoms, physicians consider escalating the treatment regimen to a triple combination of LAMA + LABA, LAMA or ICS/LABA [10,11,12,13]. Uncontrolled symptoms can lead to hospitalization or emergency room visits, and the cost of triple therapy is higher than that of treatment with LAMA or ICS/LABA alone [14,15,16,17].

We aimed to evaluate the incidence of triple combination therapy and the escalation time from the initiation of LAMA versus ICS/LABA treatment. Additionally, we identified the factors related to triple combination therapy initiation.

## 2. Materials and Methods

### 2.1. Study Design and Population

We conducted a retrospective, observational cohort study utilizing data from the National Health Insurance Service of South Korea from 1 January 2002 to 30 April 2016. The National Health Insurance Service in South Korea runs a single-payer system based on the billing records of health care providers and covers approximately 98% of the population of South Korea.

The selection period for eligible subjects was between January 2005 and April 2015. The index date was the first prescription date of LAMA or ICS/LABA during the selection period. The baseline period was defined as 36 months leading up to the index date to observe patients’ comorbidities and drug history. We enrolled long-acting inhaler–naïve patients in this study. The selected patients were followed up for a minimum of 12 months and a maximum of 136 months from the index date and were observed until the patient discontinued the drug, death, or the study period ended.

Patients aged ≥ 55 years, diagnosed with an International Classification of Diseases, Tenth Edition (ICD-10) code for COPD (J43–J44), and prescribed LAMA monotherapy or ICS/LABA fixed-dose combination during the selection period were included. We included patients with good adherence defined by the medical possession rate > 80%. Patients exposed to LAMA or ICS/LABA for a <80% medical possession rate in the first 12-month period from the index date were excluded, as they were considered unexposed to the study drugs. During the baseline period, the following factors were also assessed: history of inhaler use (ICS/LABA, LABA/LAMA, and LAMA), leukotriene receptor antagonist use, underlying significant lung diseases such as lung cancer or interstitial lung disease, lung transplantation, COPD exacerbation, and the modified Charlson Comorbidity Index. Patients with a history of using long-acting bronchodilators including ipratropium bromide, leukotriene receptor antagonist, ICS, LABA/LABA fixed-dose combination, LAMA, or ICS/LABA during the baseline period were excluded. Finally, we excluded subjects who were re-prescribed ICS, LABA, and LAMA within 30 days after the index date (Figure 1).

### 2.2. Covariates

Demographic characteristics included age, sex, and economic status (considered as income quartiles as of the index date year). Potential confounders included a history of COPD exacerbation, asthma, pneumonia, and other comorbidities. A history of COPD exacerbation was identified as ≥2 moderate exacerbations or ≥1 severe exacerbation 12 months before the index date. We defined moderate exacerbation if the patient was prescribed systemic steroids and/or antibiotics. Severe exacerbation was defined as ≥1 emergency room visit in the inpatient claims due to worsening COPD and with systemic steroids and/or antibiotics prescription. A history of asthma was identified as ≥1 claim, with ICD-10 codes for asthma recorded as the primary and fourth secondary diagnosis in the outpatient claims, or as any diagnosis in inpatient claims during the 36 months before the index date. A history of pneumonia was identified with ≥1 outpatient claims with ICD-10 codes for pneumonia recorded as any diagnosis in inpatient claims or primary and fourth secondary diagnosis in outpatient claims, a diagnostic test code for chest X-ray or chest computed tomography scan, and antibiotics prescription during the following period after the index date. Comorbidities were determined with predefined diagnosis (ICD-10) coded for relevant diseases recorded as primary and fourth secondary diagnosis during the 12 months before the index date, and the modified Charlson Comorbidity Index was used to describe a patient’s comorbidity based on predefined relevant diagnoses (ICD-10 codes) recorded during the baseline period of 12 months.

### 2.3. Outcomes

The primary outcome was the escalation time to triple therapy in patients with COPD initially treated with LAMA or ICS/LABA. Initiating triple combination therapy of ICS, LABA, and LAMA was defined by outpatient or inpatient claims with a combination of ICS, LABA, and LAMA prescription. Additional outcomes included risk factors predisposing to triple combination therapy in those initially treated with ICS/LABA compared with LAMA.

### 2.4. Statistical Analysis

We performed propensity score (PS) matching in a 1:1 ratio to balance the demographics and baseline characteristics to minimize confounding factors between the LAMA and ICS/LABA groups. Matching factors included age, sex, economic status, history of COPD exacerbation, history of asthma, history of pneumonia, modified Charlson Comorbidity Index, and index year. The continuous variables were analyzed using *t*-tests or Wilcoxon rank sum tests, and the categorical variables were analyzed using chi-square tests to compare baseline characteristics between the treatment groups. The incidence of triple combination therapy is presented as the incidence per 1000 person-years with corresponding 95% confidence intervals, the number of patients with the event, and the time to the first event from the index date. To compare risk between the two study groups, we investigated the hazard ratios and 95% confidence intervals, and conducted a Cox regression analysis. We assessed the survival probability of time to triple combination therapy using Kaplan–Meier survival curve analysis. A logistic regression analysis was performed on identified patients with COPD with LAMA and ICS/LABA exposure as the dependent variables and the listed baseline characteristics as the independent variables to identify risk factors of escalation to triple combination therapy. All statistical tests were two-sided, with a significance level of *p* < 0.05. Statistical analysis was performed using SAS version 9.4 (SAS Institute, Cary, NC, USA) via SAS Enterprise Guide version 6.1 (SAS Institute, Cary, NC, USA). 

### 2.5. Ethics Approval

This study protocol conformed to the ethical guidelines of the 1975 Declaration of Helsinki and was approved by the Institutional Review Board of Konkuk University Medical Center (Institutional Review Board No.: KUMC2020-06-013). Informed consent was waived because only de-identified database entries were accessed for analytical purposes.

## 3. Results

### 3.1. Baseline Characteristics

A total of 9284 patients were defined as the crude population; 6352 (68.4%) and 2932 (31.6%) patients were classified in the LAMA and ICS/LABA treatment groups, respectively (Appendix A). Table 1 shows the PS-matched population in a 1:1 ratio. In the PS-matched population, 2444 patients were included in the LAMA and the ICS/LABA groups. The total observation time was longer in the LAMA group than in the ICS/LABA group (787.0 ± 639.5 vs. 716.5 ± 628.4 days, *p* < 0.001) and other baseline characteristics between the two groups were evenly matched (Table 1).

### 3.2. Time to Triple Combination Therapy

The incidence of triple combination therapy for PS-matched patients was 81.0/1000 and 139.8/1000 person-years, respectively, in patients in the LAMA and ICS/LABA groups (*p* < 0.001). Patients treated with ICS/LABA were more likely to experience escalation to triple combination therapy than those treated with LAMA (383 events in the LAMA group vs. 518 events in the ICS/LABA group, *p* = 0.09)

Patients in the ICS/LABA group initiated triple therapy 74 days earlier than those in the LAMA group (281 days vs. 207 days, respectively, in the PS-matched population, *p* = 0.03). The finding that the ICS/LABA group initiated triple therapy faster than the LAMA group remains the same in the subgroup analysis by age, sex, history of COPD exacerbation, and asthma history. For all age groups, both sexes, and regardless of history of COPD exacerbation or asthma, patients treated with ICS/LABA showed an earlier initiation of triple therapy than those treated with LAMA (Table 2, Figure 2).

### 3.3. Risk Factors of Triple Combination Therapy

Various stratified analyses were performed but did not alter the results that LAMA use is associated with a decreased risk of triple combination therapy regardless of age, history of COPD exacerbation, asthma, and pneumonia compared with ICS/LABA use (Appendix A and Figure 3). The benefit of LAMA for male patients was more significant than for female patients (*p* for interaction = 0.02, hazard ratio = 1.75, 95%, confidence interval = 1.51–2.02). Multivariate analysis also supported this result (Table 3), which shows that using ICS/LABA treatment is a significant risk factor for escalation to triple therapy compared with LAMA treatment among COPD patients.

## 4. Discussion

This real-world nationwide study investigated the incidence and risk factors for triple combination therapy initiation in patients with COPD prescribed either LAMA or ICS/LABA treatment as initial therapy. Our study had two major findings. First, those initially treated with LAMA had a lower incidence for triple therapy initiation and a longer mean time to start triple therapy than those initially treated with ICS/LABA. Second, ICS/LABA prescription as an initial treatment and male sex increased the risk of escalation to triple combination therapy.

In COPD treatment, patients with a high risk of frequent exacerbations are more likely to progress to triple combination therapy [11,12,13,14,15,16,17,18] and bronchodilators such as LAMA or LABA play a crucial role in controlling the disease [19,20,21]. In our study, the incidence of escalation to triple inhaled therapy from initial LAMA use was 81.0 per 1000 PY, lower than the rate of 139.8 per 1000 PY found in the ICS/LABA group in the PS-matched sample. In addition, regarding the escalation time to triple combination therapy, we confirmed that escalation time to triple therapy was longer in the LAMA group than in the ICS/LABA group. These findings were observed regardless of age, sex, prior COPD exacerbation, and history of asthma. In a previous study by Suissa et al., ICS/LABA treatment showed clinical efficacy in patients with high eosinophil counts and resulted in a higher risk of pneumonia than LAMA treatment [22]. If patients had a history of exacerbation or asthma and were prescribed LAMA as their initial COPD treatment, their disease was likely to be well maintained with LAMA without progression to triple therapy. Therefore, ICS for step-up treatment should not be solely added based on a patient’s history of asthma, and blood eosinophil counts should be considered [7,8,9,10,11,12,13,14,15,16,17,18,19,20,21,22,23].

A recent study by Quint et al. reported that the initiation of dual bronchodilator treatment is superior to ICS/LABA in terms of the escalation time to triple therapy [24]. This study also showed that LAMA/LABA treatment was associated with a reduced risk of COPD exacerbations compared with ICS/LABA treatment, consistent with the previous findings of large COPD cohort studies [19,20,21]. Based on these studies, we found that ICS/LABA treatment is associated with an increased risk of pneumonia or exacerbation compared with LAMA treatment, and that the incidence of escalation to triple therapy could be consequently predicted.

We identified several risk factors for the initiation of triple combination therapy. The primary predictor for triple therapy initiation was ICS/LABA as the initial treatment and male sex. This result is consistent with those of previous studies, in which older males and a prescription of ICS/LABA fixed-dose combination at diagnosis were strong predictors of triple therapy initiation [23,24,25]. Interestingly, regardless of the history of asthma, ICS/LABA group had a higher risk of escalation to triple therapy than those with LAMA, and a history of asthma was not associated with triple therapy after adjustment in the PS-matched group. Therefore, we should pay attention to prescribing ICS/LABA only because patients with COPD have a history of asthma or are diagnosed with asthma–COPD overlap syndrome.

A history of ≥2 previous exacerbations is a strong predictor of COPD exacerbation [25,26,27], but was not associated with triple therapy progression in our study. This finding may be due to the exclusion of patients who were prescribed triple inhalers from the index date to 30 days after that date. Indeed, we demonstrate that the use of LAMA treatment delays the initiation of triple therapy. Again, these findings were in line with recent real-world studies comparing the escalation time to triple therapy from initiation between LAMA/LABA and ICS/LABA treatment. Notably, current COPD recommendations highlight the use of triple therapy as a follow-up treatment only when patients experience acute exacerbation and have a blood eosinophil count of >300/μL.

This study has several strengths. Unlike other large-scale observational studies, we rigorously selected patients, that is, we excluded patients who were exposed to ICS/LABA or LAMA for <12 months or had a medical possession rate of <80% and performed PS matching, providing a level of evidence comparable to a randomized controlled trial. In addition, this study included a large sample size in each group, and the data came from a real-world, nationwide setting. Finally, we were able to capture ICS/LABA or LAMA users nationwide as well as their triple combination therapy events over 5 years.

However, this study also has several limitations. First, our study was conducted using nationwide health insurance claims data, so pulmonary function test results and blood eosinophil counts were not available. In our study, COPD diagnosis was defined by an operational definition used in previous papers [28,29,30]. Second, medication adherence was not assessed objectively as with other retrospective claim data studies. Hence, only patients with a reasonable adherence rate of >80% medical possession was eligible for this study [23]. We tried to identify long-acting inhaler–naïve patients and confirm the step-up to triple combination inhaler use in patients treated with LAMA or ICS/LABA as much as possible, so we included 36 months before the index date as the baseline period. Third, this study excluded patients who were re-prescribed triple combination therapy within 30 days after initiating the index drug. These exclusion criteria were applied to minimize the number of patients who should have been prescribed either LAMA or ICS/LABA at initiation. However, this is a critical limitation of the study as it may reflect only a subset of the entire COPD population. Last, in our study using claims data, it was difficult to distinguish the prescription of ICS/LABA among patients with COPD and asthma because we did not exclude patients who had a diagnostic code for asthma. However, we excluded patients who could have had asthma (defined as prescribed with SABA or LTRA or aged < 55 years) as much as possible to reduce the number of patients treated with ICS/LABA for asthma. In addition, we attempted to reduce this bias through the PS matching method.

This study found that initial use of LAMA therapy to treat COPD was associated with a reduced rate of escalation to triple therapy compared with that of ICS/LABA.

## Figures and Tables

**Figure 1 jpm-11-01325-f001:**
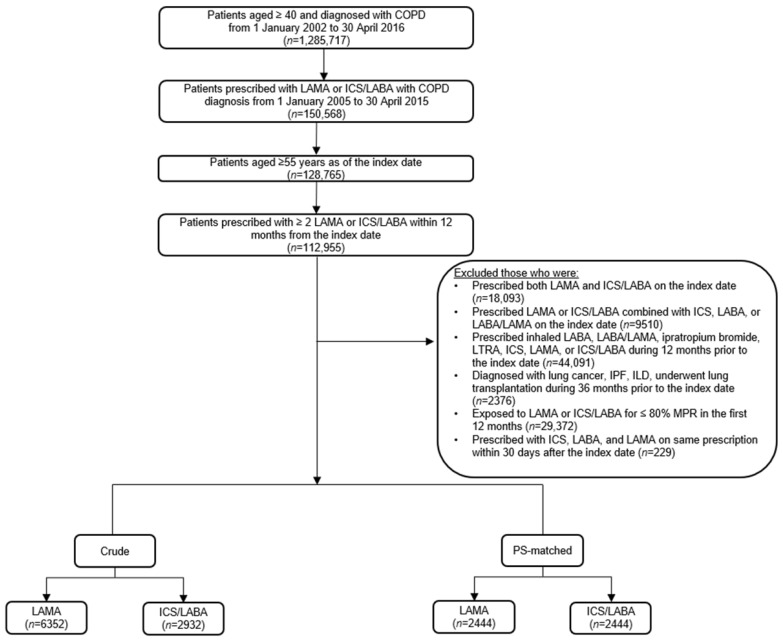
Flowchart of the study population. LAMA, long-acting muscarinic antagonist; ICS/LABA, inhaled corticosteroid plus long-acting beta-2 agonist; COPD, chronic obstructive pulmonary disease; MPR, medical possession rate; LTRA, leukotriene receptor antagonist; ILD, interstitial lung disease.

**Figure 2 jpm-11-01325-f002:**
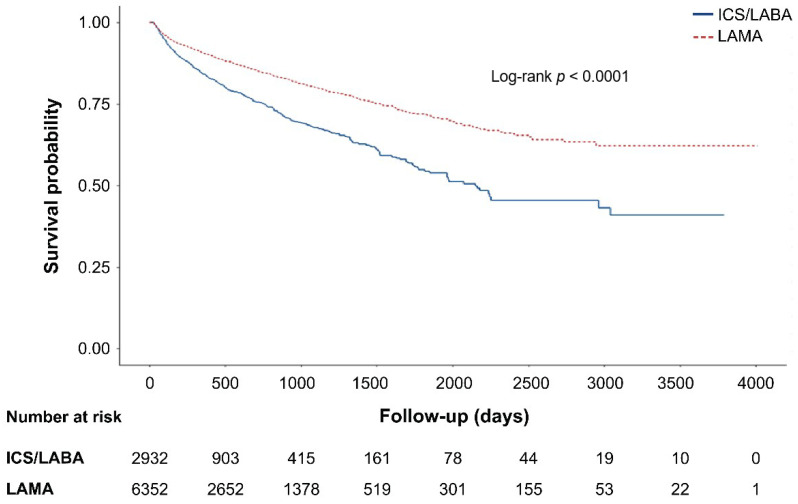
Survival curve of patients under two treatments (long-acting muscarinic antagonist (LAMA) vs. inhaled corticosteroids/long-acting beta2 agonist (ICS/LABA) until escalation to triple combination therapy.

**Figure 3 jpm-11-01325-f003:**
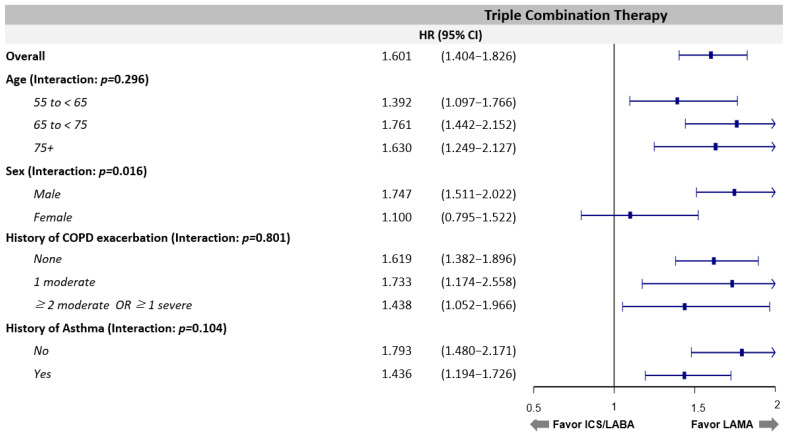
Forest plot comparing triple combination therapy initiation between ICS/LABA and LAMA.

**Table 1 jpm-11-01325-t001:** Baseline characteristics of the propensity score-matched study population.

Baseline Characteristics	All (*n* = 4888)	LAMA (*n* = 2444)	ICS/LABA (*n* = 2444)	*p* Value
Observational period, days	733.92 ± 634.14	801.46 ± 698.29	666.39 ± 554.84	<0.001
Age, years	69.7 ± 7.9	69.6 ± 7.9	69.7 ± 7.9	0.53
55 to <65	1396 (29)	697 (29)	699 (29)	
65 to <75	2106 (43)	1068 (44)	1038 (42)	
75+	1386 (28)	679 (28)	707 (29)	
Sex, male	3676 (75)	1854 (76)	1822 (75)	0.29
Income level				
1st quartile	768 (16)	376 (15)	392 (16)	0.77
2nd quartile	634 (13)	311 (13)	323 (13)	
3rd quartile	935 (19)	473 (19)	462 (19)	
4th quartile	1641 (34)	838 (34)	803 (33)	
Medical aid	910 (19)	446 (18)	464 (19)	
History of COPD exacerbation				
None	3648(75)	1831 (75)	1817 (74)	0.88
1 moderate	485 (10)	87 (7)	247 (10)	
≥2 moderate OR ≥ 1 severe	755 (15)	375 (15)	380 (16)	
History of asthma				
No	2530 (52)	1263 (52)	1267 (52)	0.91
Yes	2358 (48)	1181 (48)	1177 (48)	
History of pneumonia				
No	4377 (90)	2179 (89)	2198 (90)	0.37
Yes	511 (10)	265 (11)	246 (10)	
mCCI	2.0 ± 1.8	1.9 ± 1.8	2.0 ± 1.8	0.19

Data are expressed as mean ± standard deviation or *n* (%). LAMA, long-acting muscarinic antagonist; ICS/LABA, inhaled corticosteroid plus long-acting beta-2 agonist; COPD, chronic obstructive pulmonary disease; mCCI, modified Charlson Comorbidity Index.

**Table 2 jpm-11-01325-t002:** Incidence of triple combination therapy and time to triple combination therapy in the propensity score matched population.

Baseline Characteristics	All (*n* = 4888)	LAMA (*n* = 2444)	ICS/LABA (*n* = 2444)	*p* Value
Incidence rate per 1000 person-years	106.8	81.0	139.8	<0.001
Patients with event	901	383	518	0.09
Median time to event	227	281	207	0.03
Age (years)				<0.001
55 to <65	435.3 ± 501.6	460.0 ± 558.9	415.0 ± 449.9	
65 to <75	397.96 ± 461.1	479.63 ± 532.2	338.8 ± 392.6	
75+	349.41 ± 406.8	382.67 ± 381.8	327.1 ± 422.7	
Sex				
Male	406.0 ± 464.1	473.9 ± 514.8	358.2 ± 418.9	<0.001
Female	353.4 ± 450.0	349.1 ± 482.4	357.6 ± 420.3	0.001
History of COPD exacerbation				
None	395.9 ± 458.2	444.7 ± 503.6	360.0 ± 418.9	<0.001
1 moderate	416.3 ± 502.4	513.7 ± 566.9	347.61 ± 443.5	<0.0001
≥ 2 moderate OR ≥ 1 severe	391.0 ± 452.1	433.9 ± 505.9	357.6 ± 405.1	0.005
History of asthma				
No	395.6 ± 457.4	461.6 ± 497.9	351.7 ± 423.6	<0.001
Yes	399.1 ± 466.8	441.0 ± 522.0	364.8 ± 414.3	<0.001

Data are expressed as mean ± standard deviation or *n* (%). LAMA, long-acting muscarinic antagonist; ICS/LABA, inhaled corticosteroid plus long-acting beta-2 agonist; COPD, chronic obstructive pulmonary disease.

**Table 3 jpm-11-01325-t003:** Risk factors influencing triple combination escalation among the propensity score-matched population.

Variables		Univariate HR			Multivariate HR		
		HR	95% CI	*p* Value	aHR	95% CI	*p* Value
Index med	LAMA	ref.	-		ref.	-	-
	ICS/LABA	1.601	1.402–1.829	<0.001	1.632	1.428–1.864	<0.001
Age (years)	55 to <65	ref.	-		ref.	-	-
	65 to <75	0.950	0.815–1.107	0.511	1.046	0.814–1.344	0.725
	75+	0.824	0.691–0.983	0.031	1.024	0.656–1.597	0.918
Sex	Female	ref.	-		ref.	-	-
	Male	1.670	1.400–1.992	<0.001	1.699	1.422–2.031	<0.001
History of COPD exacerbation	None	ref.	-	-	ref.	-	-
	1 moderate	1.225	0.996–1.507	0.055	1.213	0.983–1.497	0.072
	≥2 moderate OR ≥1 severe	1.194	1.004–1.420	0.045	1.191	0.996–1.424	0.056
History of asthma	No	ref.	-	-	ref.	-	-
	Yes	1.133	0.994–1.291	0.061	1.146	0.998–1.317	0.053
History of pneumonia	No	ref.	-	-	-	-	-
	Yes	1.029	0.831–1.274	0.795	-	-	-
Congestive heart failure	No	ref	-	-	-	-	-
	Yes	1.002	0.811–1.238	0.988	-	-	-

Data are expressed as mean ± standard deviation or *n* (%). LAMA, long-acting muscarinic antagonist; ICS/LABA, inhaled corticosteroid plus long-acting beta-2 agonist; COPD, chronic obstructive pulmonary disease, HR, hazard ratio; CI, confidence interval; aHR, adjusted hazard ratio.

## Data Availability

The National Health Insurance Service data is an open and public data to which any researcher can request access at https://nhiss.nhis.or.kr (accessed on 23 March 2020).

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
