# Peer review of "Escalation Time to Open Triple Combination Therapy from the Initiation of LAMA versus ICS/LABA in COPD Management: Findings from Comparing the Incidence of Tiotropium and ICS/LABA in Real-World Use in South Korea (CITRUS) Study"

_jpm, 2021, doi:10.3390/jpm11121325_

Round 1
Reviewer 1 Report
The aim of the paper is interesting. The number of patients and the analyze period is good.
The conclusion is very suprising and readers could used this information for citations.
Line 28-29- It would be beter to have period near to the 2020 not until 2015 in the study, a fresher period for analysis („This retrospective study included COPD patients selected from the National Health Insurance Service of South Korea from January 2005 to April 2015.“)
Line 101 – Consider includes and the level of education for the variable (without school, - primary school, secondary school, faculty) („Demographic characteristics included age, sex, and economic status (considered as ncome quartiles as of the year of the index date)“).
Line 115 – Also, it is very interesting to put as variable the most common comorbitiy to evaluate for escalating to the triple therapy
Tables are readable.
Conclusions short and effectively
Author Response
Dear reviewer,
Please find the response to your review as attached.
Thank you.

Reviewer 2 Report
The subject is very interesting. the article is well structured. The results cover the aims of the study. The conclusions are supported by the results. However a few observations:
The introduction: elaborate more about the reason to escalation to triple therapy, maybe cite few studies.
line 58- 61 : the phrase is very difficult to read. Rephrase it
Discussion: Should be more detailed. The references could be more updated there are recent papers that should appear in the discussion section
The authors should refrain from statements like: "we clearly demonstrate" (lin e256)
The limitations of the study are very well emphasized.
Author Response
Dear Reviewer,
Ref: jpm-1446922
Article Title: Escalation time to open triple combination therapy from the initiation of LAMA versus ICS/LABA in COPD management: findings from the Comparing the Incidence of Tiotropium and ICS/LABA in Real-World Use in South Korea (CITRUS) study
We would like to thank all of the editors and reviewers for helping us make a better revision. We revised our manuscript according to the comments and recommendations of the reviewers. We have highlighted all changes in the revised manuscript in underlined red font. Below, we have included an itemized series of responses to the comments of the reviewers. We've also uploaded the file as for your reference.
- The introduction: elaborate more about the reason to escalation to triple therapy, maybe cite few studies.
Response) Thank you for your valuable comment. As you recommended, we elaborate the reason to escalation to triple therapy (unsatisfactory improvement of symptoms, or reduced lung function) by citing more studies.
“When patients with COPD experience frequent exacerbations, reduced lung function or worsening symptoms, physicians consider escalating the treatment regimen to a triple combination of LAMA+LABA or LAMA or ICS/LABA.10-13.”
- line 58- 61 : the phrase is very difficult to read. Rephrase it
Response) Thank you for your thoughtful comments. We agree with your opinion, so we rephrase it as follow;
“We aimed to evaluate the incidence of triple combination therapy and the escalation time to open triple combination therapy from the initiation of LAMA versus ICS/LABA treatment. Additionally, we identified the factors related to triple combination therapy initiation.”
- Discussion: Should be more detailed. The references could be more updated there are recent papers that should appear in the discussion section
Response) Thank you for your comments. We cited paper more recent papers in the discussion section.
- The authors should refrain from statements like: "we clearly demonstrate" (line 256)
Response) We appreciate the reviewer’s valuable advice, and we agree with that statement “we clearly demonstrate” should be refrain. So, we deleted “clearly”.
- The limitations of the study are very well emphasized.
Response) Thank you for your comments.